biophysics/differential equations/statistical physics

cellular automaton, absorbing phase transition, tissue, generalized epidemic process

**Author for correspondence:**
Yuting Lou
e-mail: chelinqueen@hotmail.com

# Homeostasis and systematic ageing as non-equilibrium phase transitions in computational multicellular organizations

Yuting Lou[1], Ao Chen[2], Erika Yoshida[3] and Yu Chen[1]

[1]SCS Laboratory, Department of Human and Environmental Engineering, Graduate School of Frontier Sciences, The University of Tokyo, 277-8561 Chiba, Japan
[2]Department of Physics, Fudan University, Shanghai, People's Republic of China
[3]Department of Systems Innovation, Faculty of Engineering, The University of Tokyo, 111-8564 Tokyo, Japan

YL, 0000-0002-1634-1673

Being a fatal threat to life, the breakdown of homeostasis in tissues is believed to involve multiscale factors ranging from the accumulation of genetic damages to the deregulation of metabolic processes. Here, we present a prototypical multicellular homeostasis model in the form of a two-dimensional stochastic cellular automaton with three cellular states, cell division, cell death and cell cycle arrest, of which the state-updating rules are based on fundamental cell biology. Despite the simplicity, this model illustrates how multicellular organizations can develop into diverse homeostatic patterns with distinct morphologies, turnover rates and lifespans without considering genetic, metabolic or other exogenous variations. Through mean-field analysis and Monte–Carlo simulations, those homeostatic states are found to be classified into extinctive, proliferative and degenerative phases, whereas healthy multicellular organizations evolve from proliferative to degenerative phases over a long time, undergoing a systematic ageing akin to a transition into an absorbing state in non-equilibrium physical systems. It is suggested that the collapse of homeostasis at the multicellular level may originate from the fundamental nature of cell biology regarding the physics of some non-equilibrium processes instead of subcellular details.

## 1. Introduction

Multicellular organizations in living things are maintained in homeostasis, which refers to a dynamical balance of loss and gain of cells that maintains the functions and the integrity of a

system. There exists a diversity of homeostatic patterns in respect of morphology, functionality, cell subpopulation structure, turnover rate, and lifespan in various tissues [1,2]. Such diversity is supposed to originate from numerous intra-, inter- and extra-cellular factors, the discoordination of which can lead to age-related diseases such as degenerative ageing and cancer [3–6]. Throughout the decades, molecular biology has been placed as the priority to build understanding on homeostasis from each detail, yet still leaving many details unclarifiable. As an alternative, multiscale computational modelling thence becomes an effective tool to test more hypotheses and helps elucidate the mechanisms of multicellular phenomena from a systematic perspective [7–12]. Previous studies on epithelial embryogenesis have shown that a small variation in the cell sensitivity to extrinsic and intrinsic signals may lead to the formation of different homeostatic patterns, including degenerative and tumorigenic ones [13,14]. Longtime simulations of these systems suggested that transitions from healthy homeostasis to abnormal ones may spontaneously occur without external or internal variations [15] and the ageing process shows features of dynamical scaling as the physical ageing does in complex matters like glasses [16]. The most interesting finding lies in that the nature of homeostatic diversity and quasi-stability seems to depend sensitively on some factors such as the response to the extracellular matrix (ECM) [13,17] while insensitively on others such as the hydrodynamics of cell membranes and of the ECM [15].[1]

To elucidate the governing factor of diverse and quasi-stable homeostasis, above-mentioned models are not suitable because of their complicated parametrization. Models with fewer assumptions and mathematical analysability thereby can serve as feasible platforms to abstract a neat theory. In fact, some cell-based models such as cellular Potts models [18,19], lattice-gas cellular automata [20,21], and centre-based single-cell models [22,23] have been extensively applied in cell sorting, tissue morphogenesis and tumour growth, etc., (see a review in [8,24,25]), and some simple stochastic processes like the critical birth–death process [26–28], the voter model [29] or the combined version [30], have also been proposed for describing the 'differentiation-or-not' decision-making of progenitor cells to explain the lineage-tracing data of *in vivo* epidermal tissues [27,31]. It is also analytically shown that, with density-dependent autoregulating mechanisms [32], these models can also exhibit a metastable homeostatic state with a fixed cell number as well as a stable state of an expanding cell population; hence a hypothesis that cancer occurs as a rare event of the system escaping homeostatic basin of attraction is proposed. These previous studies manifested the power of simpler models in explaining macroscopic phenomena of interest to revealing skeletal mechanisms irrelevant to details.

For our purpose to address the critical mechanism underlying the formation of diverse homeostatic patterns with the healthiest ones ageing at the multicellular level, we will present a prototypical model in the form of a two-dimensional stochastic cellular automaton (SCA) [33,34], where the dynamics of elementary components of the system (i.e. a cell) is discretized in time and space. To avoid redundant details, we consider only three fundamental phenotypic states of the modelled cell: proliferative, dead or quiescent (cell cycle arrest), without distinguishing differentiated and stem/progenitor cells. The cell switches its state, or phenotype, at a rate depending on the numbers of neighbours with respect to the three phenotypes. These state-updating rules are set on the most acknowledged cell biology regarding cell division, death and cell cycle. Other sophisticated details are aggregately modelled by the stochasticity involved in the implementation of state-updating. To exclude any influence from the genetic, metabolic and environmental turbulence, cells are assigned with homogeneous rules.

With such simple settings, mathematical analyses could be easily achieved using mean-field techniques to have an overall understanding on the model behaviour; meanwhile, extensive Monte–Carlo simulations for statistical analyses reveal the complete phase diagrams for both infinite and finite space. It is found that the system can form into homeostasis with distinct geometries, phenotypic subpopulation structures, turnover rates and lifespans just as found in complicated multiscale models [13,15]. All these homeostatic states can be classified on the phase diagram into extinctive, proliferative or degenerative phases. In a finite space, the proliferative phases split into three subphases including tumourigenic and healthy homeostasis, the latter of which, standing on the critical boundary between tumourigenic and degenerative phases, has a large spectrum of lifespans with quasi-stable cyclic dynamics before collapsing into degenerative phases. Should more biological details be considered to entail spatio-temporal heterogeneity in the system, the healthy proliferative phases in the parameter space expand further. This quasi-stable nature of healthy homeostasis

---

[1]However, such fluid mechanics are very important to reproduce exact multicellular architectures that are comparable to experimental observations.

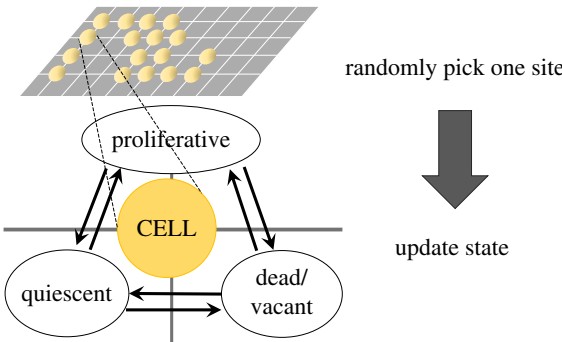

**Figure 1.** An SCA on a two-dimensional regular lattice with three states of a site: occupied by a proliferative cell (1), by a quiescent cell ($-1$), or vacant (0). Note that asynchronous updating is adopted. The arrows between pairs of three states represent a change of state with some probability based on fundamental cell biology (table 1).

originates from its standing on the critical boundary between the proliferative and degenerative phases, hence the transitions from healthy to degenerative phase indeed recalls the absorbing phase transitions in non-equilibrium physical systems [35–37]. This implies that ageing at the multicellular level could originate from some innate symmetry-breaking in the non-equilibrium intercellular dynamics, akin to physical ageing in the non-equilibrium physical systems [16,38] even without consideration of genetic, metabolic deregulation or exogenous variations.

The outline of this article is as follows: model details and some basic output are introduced in §2; the core mechanism underlying the model behaviours are elucidated through mean-field analysis in §3 and through analysis from a statistical mechanics perspective in §4. Section 5 gives some discussions on the possible model extensions towards more specific biological systems and on some well-studied non-equilibrium stochastic processes most relevant to our model; §6 concludes this work.

# 2. Model

## 2.1. Basic description

SCA are Markov processes defined on a product space $\prod_{x\in S} \Psi(x)$, where $\Psi(x)$ is a collection of states of elementary entity at position $x$ in space **S**. Among the most studied SCAs are the stochastic Ising model for studying magnetism, the voter model for the competition for the majority and opinion formation, the contact process for epidemic spreading and specific chemical reactions, and so forth (see reviews in [34,39]). Our model is defined as a Markov process with state space $\{1, 0, -1\}^{\mathbb{Z}^2}$, where '1' represents proliferative, '0' dead or vacant, and '$-1$' quiescent state of each entity, which is a site occupying at most one cell. Cells in reality can have many 'states', or more strictly speaking, 'phenotypes', as a consequence of being at some metastable status of the endogenous network of signalling pathways that decide the level of expression of proteins [40,41]. For studying homeostasis, proliferation, quiescence and death are three most crucial states first to be considered. Cell quiescence is a state where the cell exits the cell cycle to stop proliferation while living with low metabolic demand, and this state dominates in most of tissues [42,43]. Cell death in our model is just an instant elimination of the cell, leaving the site vacant. To keep model minimal, no cell migration or other mechanics are involved.

The space is a two dimensional regular lattice with periodic boundary conditions. At each time step $t$, the states of sites in the system are updated based on a set of homogeneous rules. Thus no discrimination on stem cells, progenitor cells or different types of differentiated cells is made. In all the Monte–Carlo simulations below, asynchronous updating is implemented (i.e. the sequence of updating sites is randomly generated at each $t$). Figure 1 shows the basic construction of the model.

The updating rules are defined as a set of probabilities of transitions from one state to another $p_{i\to j}$, $i, j \in \{1, 0, -1\}$ for each site. Also note that for each site with a state $i$, the sum of transition probabilities $\sum_j p_{i\to j}$ should be kept as unity, hence only six independent probabilities exist. These probabilities could be qualitatively postulated from fundamental cell biology. For instance, as a universal principle for a living entity, the state '$-1$', i.e. the quiescence of cell cycle, occurs only in a living cell, hence

**Table 1.** Transition matrix for updating state at each site. (Note: the probability $p_{i \to i}$ (shown as asterisk (*)) keeps the sum of all three elements in each row as 1. Growth rate $g$ is set as 1.)

| $p_{i \to j}$ \ $j$ <br> $i$ | 1 | 0 | $-1$ |
|---|---|---|---|
| 1 | * | $d$ | $\dfrac{a \sum_{n \in \mathbf{N}} \delta(\lvert \eta(n) \rvert, 1)}{\lvert \mathbf{N} \rvert}$ |
| 0 | $\dfrac{g \sum_{n \in \mathbf{N}} \delta(\eta(n), 1)}{\lvert \mathbf{N} \rvert}$ | * | |
| $-1$ | $\dfrac{r \sum_{n \in \mathbf{N}} \delta(\eta(n), 0) \cdot \sum_{n \in \mathbf{N}} \delta(\eta(n), 1)}{\lvert \mathbf{N} \rvert^2}$ | $\dfrac{d \sum_{n \in \mathbf{N}} \delta(\eta(n), 0)}{\lvert \mathbf{N} \rvert}$ | * |

$p_{0 \to -1}$ must be zero. The remaining five transition probabilities will be specified one by one in the next subsection.

## 2.2. Updating rules

Before introducing any equations, let us first clarify some notations: (i) $\eta(x) \in \{-1, 0, 1\}$ is the site state at a given site $x$; (ii) $\mathbf{N}_x$ is the set of Von Neumann neighbouring sites[2] of $x$. For convenience, all $x$ will be dropped in the equations below and in table 1; $\lvert \mathbf{N} \rvert$ denotes the size of the neighbour set, which is 4 in our model; (iii) $\delta(i, j)$ is the Kronecker delta equal to 0 when $i \neq j$ and 1 when $i = j$.

(a) If the site state is '0' (vacant): the probability for a vacant site to become occupied by a proliferative cell is proportional to the number of neighbouring proliferative cells:

$$p_{0 \to 1} \propto \frac{\sum_{n \in \mathbf{N}} \delta(\eta(n), 1)}{\lvert \mathbf{N} \rvert}. \qquad (2.1)$$

By contrast, there is a zero probability for a vacant site to become a quiescent state, i.e. $p_{0 \to -1} = 0$.

(b) If the site state is '1' (occupied by a proliferative cell): the probability for a proliferative cell to become quiescent increases with the number of neighbouring living cells as

$$p_{1 \to -1} \propto \frac{\sum_{n \in \mathbf{N}} \delta(\lvert \eta(n) \rvert, 1)}{\lvert \mathbf{N} \rvert}. \qquad (2.2)$$

This growth inhibition caused by the local congestion of cells models the so-called 'contact inhibition' observed in many multi-cell experiments [44–46]. The underlying biology may involve the accumulation of cell–cell adhesion and cell–ECM adhesion that block the growth signals to cells.

The probability for a proliferative cell to die is predefined as $p_{1 \to 0} = d$, which is uniform and constant. Cell death can generally be categorized into apoptosis and necrosis [47,48], the former of which is a self-programmed suicide owing to sensing sufficient death-inducing signals, while the latter is an unprogrammed death owing to external damages, such as hypoxia, injury, heat and radiation. No distinction between the two is made in our model and the death probability $d$ is a parameter aggregating all the factors adverse to cell survival, and we will call it 'background death probability' hereafter.

(c) If the site state is '$-1$' (occupied by a quiescent cell): basically, the quiescent cells are rather insensitive to signals because of a strong adherence of cell membranes to ECMs and neighbours, which leaves fewer receptor domains for sensing other cues. Hence, a prerequisite for quiescent cells to be reactivated is the loss of contact inhibition, i.e. the decrease in ECM and intercellular adhesion. In our model, this is modelled by the appearance of local vacancy of sites. Therefore, both $p_{-1 \to 1}$ and $p_{-1 \to 0}$ have to increase with $\sum_{n \in \mathbf{N}} \delta(\eta(n), 0)$.

[2]Complicated definitions of neighbour set could be intended for more specific biological problems. For our purpose, the Von Neumann neighbour set is sufficient to represent the local range of intercellular interactions.

However, the other conditions for cells restarting growth is sensing sufficient growth factors, which are mostly secreted by local proliferative cells through autocrine and paracrine signalling [49–51]. Accordingly, some cells may never have chance to restore if they are surrounded by other quiescent cells. Combining the two aspects, the probability for a cell to transition from quiescence to proliferation is

$$p_{-1 \to 1} \propto \frac{\sum_{n \in \mathbf{N}} \delta(\eta(n), 0)}{|\mathbf{N}|} \cdot \frac{\sum_{n \in \mathbf{N}} \delta(\eta(n), 1)}{|\mathbf{N}|}, \tag{2.3}$$

where the first term stands for the loss of contact inhibition, and the second term for the intensity of growth signals.

Similarly, the death probability for a quiescent cell should depend on another factor: the background death probability $d$ and thus the probability for a cell to transition from quiescence to death is

$$p_{-1 \to 0} \propto d \cdot \frac{\sum_{n \in \mathbf{N}} \delta(\eta(n), 0)}{|\mathbf{N}|}. \tag{2.4}$$

As one may observe, the regrowth probability $p_{-1 \to 1}$ is smaller than the birth probability $p_{0 \to 1}$ and the probability $p_{-1 \to 0}$ is smaller than the background death probability $p_{1 \to 0}$, agreeing with the fact that the quiescent cells are less sensitive to cues than proliferative cells.

All the updating rules are summarized in table 1. There are four parameters: the growth rate $g$, the regrowth ability $r$, the tendency of cell cycle arrest $a$ and the death rate $d$. Without the loss of generality, the growth rate $g$ is set as unity and only three independent parameters $r, d, a$ govern the system dynamics. With $g = 1$, the single-cell division rate is $1/4$ in one time step, hence the doubling time of a cell is effectively $\ln 2/0.25 \approx 2.77$ time steps in our model. Considering that single-cell doubling times for most cells *in vitro* are nearly 24 h [52,53], one time step in our model roughly corresponds to 8.66 h in real time. Remember that the ranges of $r, d$ and $a$ must meet the constraints which guarantee that the sum of all elements in each row in the table is unity.

## 2.3. Output

Before analysing the model behaviours, we first present several visualized outputs from Monte Carlo simulations with varying parameters $r, d, a$ for grasping some intuition (the code for the Monte Carlo simulations is included in the electronic supplementary material). The initial condition is to seed one proliferative cell in the space, which is a two-dimensional lattice with a periodic boundary condition. With this initial condition, the output dynamics mimic the embryogenic development of multicellular organisms, yet the final homeostatic patterns of this developmental process are insensitive to initial conditions. The linear size of system $L$ is set as 100 in the visualized output (figure 2). Several different patterns could be identified straightforwardly through the evolution of the distribution of proliferative cells and quiescent cells in space. Figure 2 shows seven typical patterns, including quick extinction (figure 2a), slow extinction (figure 2b), scattered growth (figure 2c), compact growth (figure 2d), oscillatory growth (figure 2e), oscillatory degeneration (figure 2f), quick degeneration (figure 2g). In each subfigure, the right-hand plot exhibits the time evolution of the densities of proliferative cells (red) and quiescent cells (grey).

Basically, the seven homeostatic patterns could be classified into three major types, namely extinctive, proliferative and degenerative. The extinctive patterns fail to form an expanding cell cluster and the system struggles to survive for a time with a large spectrum (figure 2a,b). The proliferative patterns include three sub-patterns of growth: one with many small clusters scattered in the space (figure 2c), one with a compact expanding cluster (figure 2d), and one with an oscillatory shift between the scattered and the compact growth. In the electronic supplementary material, one can see the spatio-temporal oscillations as the wave propagations of the growth signals (red dots). The degenerative patterns are those able to form expanding clusters at early stages but undergoing irrevocable arrest of system dynamics (owing to the domination of quiescent cells), wherein some systems may struggle to survive in an oscillatory pattern over an extremely long time (figure 2g) and others degenerate quickly (figure 2e).

It is notable that the transition from the oscillatory growth to oscillatory degeneration seems to sensitively depend on the parameters ($d$ from 0.05 to 0.04) and the oscillatory degeneration prior to the final collapse of homeostasis looks similar to the oscillatory growth over a long time period. Meanwhile, zooming into the compositional percentage of proliferative and quiescent cells (see the

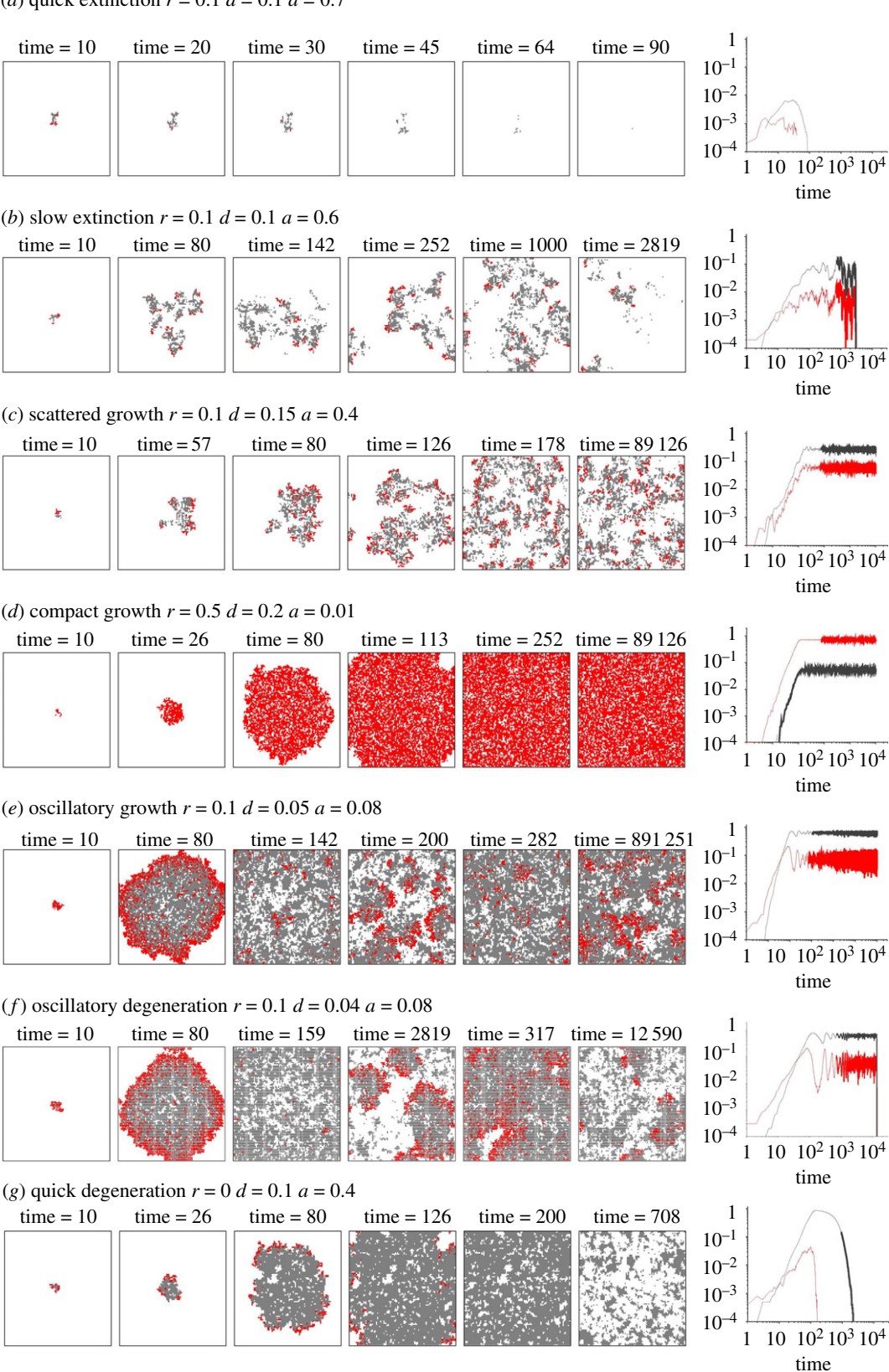

**Figure 2.** $(a-g)$ Formation of diverse homeostatic patterns. Red dots represent proliferative cells and grey quiescent cells. The right-hand column exhibits the time evolution of the densities of proliferative cells (red line) and quiescent cells (grey).

right-hand plots in figure 2$e$,$f$), one can find that nearly 90% of the total cells are quiescent cells with very few proliferative cells to compensate for cell death. These are the features of healthiest homeostasis in tissues as quiescent cells help maintain tissue architectures and function with low metabolic

**Table 2.** Lifespans and turnover rates for the sample homeostatic patterns in figure 2.

| homeostatic pattern | lifespan (step) | turnover rate/step |
| --- | --- | --- |
| (a) quick extinction | $9.5 \times 10^2$ | |
| (b) slow extinction | $3.1 \times 10^3$ | $0.045 \pm 0.018$ |
| (c) scattered growth | $\infty$ | $0.073 \pm 0.009$ |
| (d) compact growth | $\infty$ | $0.185 \pm 0.004$ |
| (e) oscillatory growth | $\infty$ | $0.013 \pm 0.004$ |
| (f) oscillatory degeneration | $1.4 \times 10^4$ | $0.009 \pm 0.004$ |
| (g) quick degeneration | $2.3 \times 10^3$ | |

consumption. Other patterns like the scattered or compact growth, either featuring a fragmented morphology or an enormous number of proliferative cells,[3] lack stability in multicellular architecture and function.

Besides the geometric characteristics and the subpopulation structure, these homeostatic patterns also differ in cell turnover rate and lifespan. Cell turnover rate characterizes how fast old cells are replaced by newly born ones [54] and can be measured in our model by the ratio of newly born cells over total cell number, averaged over the survival time; the lifespan of a system is more straightforwardly measured by the time when the whole space becomes vacant or filled with quiescent cells. Table 2 shows the turnover rate and lifespan for each homeostatic pattern. In real tissues, the turnover time (the reciprocal of turnover rate) ranges widely from several days to several years [54–57] and considering that one step in our model corresponds to roughly 1/3 day (see §2), only the oscillatory growth and oscillatory degeneration patterns have turnover times comparable with the homeostasis in reality. Therefore, these two patterns are most significant to biology.

The diverse homeostatic patterns and the collapse of the healthy homeostasis (the oscillatory degeneration) by this simple model are similar to those reproduced by a previous complicated receptor dynamics model [15], implying that there exist some universal mechanisms irrelevant to details. In the next two sections, we will approach this core mechanism by mean-field analysis and phase diagrams.

# 3. Mean-field analysis

Mean-field approach has been extensively used in the stochastic models to reduce a many-body problem to a one-body problem, through which fluctuations caused by spatio-temporal heterogeneous interactions are neglected. Although the mean-field analysis may give solutions deviating from the Monte−Carlo simulations, they are feasible to give analytical description of the basic model properties.

The dynamics of the system can be characterized by the time evolution of proliferative cells density, denoted as $\rho_p(t)$ and that of quiescent cells, denoted as $\rho_q(t)$. At each time for each site $x$, the probability for finding it in a proliferative and quiescent state is $P_p(x, t)$ and $P_q(x, t)$. Thence the master equation (which governs the stochastic dynamics) of the system is

$$
\begin{cases}
\dfrac{\mathrm{d}P_p(x, t)}{\mathrm{d}t} = p_{0\to1}^x(t)(1 - P_p(x, t) - P_q(x, t)) + p_{-1\to1}^x(t)P_q(x, t) \\
\qquad - (p_{1\to-1}^x(t) + p_{1\to0}^x(t))P_p(x, t) \\
\text{and} \quad \dfrac{\mathrm{d}P_q(x, t)}{\mathrm{d}t} = p_{1\to-1}^x(t)P_p(x, t) - (p_{-1\to0}^x(t) + p_{-1\to1}^x(t))P_q(x, t),
\end{cases}
\tag{3.1}
$$

where $p_{i\to j}^x$ ($i, j \in \{1, 0, -1\}$) is the transition probability from state $i$ to state $j$ for the site $x$ (table 1). Note that $1 - P_p(x, t) - P_q(x, t)$ is the probability of site $x$ to be vacant. The terms on the right with positive signs account for the amount of gain in probability, and negative for the amount of loss. Summing up equations (3.1) over the space, the left side will approximate the number of proliferative

---

[3]In this model, these proliferative cells will not divide owing to the limited space; in real situations, tissues locally dominated by proliferative cells expand to a large size and form tumours then.

cells or quiescent cells in the limit of system size $L$ approaching infinity, i.e. $\sum_{x \in S} P_*(t) \overset{L \to \infty}{=} \rho_*(t)|S|$; hence we have

$$
\begin{cases}
\dfrac{d\rho_p(t)}{dt} \overset{L \to \infty}{=} \dfrac{1}{|S|} \sum_{x \in S} \Big( p_{0 \to 1}^x(t)(1 - P_p(x, t) - P_q(x, t)) + p_{-1 \to 1}^x(t)P_q(x, t) \\
\qquad\qquad - (p_{1 \to -1}^x(t) + p_{1 \to 0}^x(t))P_p(x, t) \Big) \\[4pt]
\text{and} \quad \dfrac{d\rho_q(t)}{dt} \overset{L \to \infty}{=} \dfrac{1}{|S|} \sum_{x \in S} \Big( p_{1 \to -1}^x(t)P_p(x, t) - (p_{-1 \to 0}^x(t) + p_{-1 \to 1}^x(t))P_q(x, t) \Big),
\end{cases}
\tag{3.2}
$$

where $|S|$ is the total site number $L^2$ in the space. In equation (3.2), the transition probability $p_{i \to j}^x(t)(i, j \in \{1, 0, -1\})$ characterizes the state-updating in local space and time (table 1). Under mean-field assumptions, these probabilities can be effectively replaced by one global quantity $p_{i \to j}^g(t)$, which is uniform across the whole system of infinitely large, hence all the neighbour set $N$ in equations (2.1)–(2.4) could be replaced by the lattice site set $S$. Thus, equation (3.2) turns into a set of two-variable nonlinear ordinary differential equations:

$$
\begin{cases}
\dfrac{d\rho_p(t)}{dt} \overset{L \to \infty}{=} (1 + r\rho_q(t))(1 - \rho_p(t) - \rho_q(t)) \\
\qquad\qquad - \Big( a(\rho_p(t) + \rho_q(t)) + d \Big)\rho_p(x, t) \\[4pt]
\text{and} \quad \dfrac{d\rho_q(t)}{dt} \overset{L \to \infty}{=} a(\rho_p(t) + \rho_q(t))\rho_p(t) \\
\qquad\qquad - (d + r\rho_p(t))(1 - \rho_p(t) - \rho_q(t))\rho_q(x, t)
\end{cases}
\tag{3.3}
$$

Let the time derivatives on the left side of equation (3.3) be zero and solve the equations to derive three fixed points $(\rho_p^*, \rho_q^*)$: $(0, 0)$, $(0,1)$ and $(\rho_{p3}, \rho_{q3})$, where

$$
\begin{cases}
\rho_{p3} = \dfrac{(1 - d)(rd - a)}{rd} + \varepsilon_p \\[4pt]
\text{and} \quad \rho_{q3} = \dfrac{a(1 - d)}{rd} - \varepsilon_q,
\end{cases}
\tag{3.4}
$$

with two correction terms $\varepsilon_p > 0$ and $\varepsilon_q > 0$.[4]

Next, one can analyse the linear stability of system dynamics near these fixed points by analysing the Jacobian matrix $J$, which is

$$
\begin{bmatrix}
\dfrac{\partial \dot\rho_p}{\partial \rho_p} & \dfrac{\partial \dot\rho_p}{\partial \rho_q} \\[6pt]
\dfrac{\partial \dot\rho_q}{\partial \rho_p} & \dfrac{\partial \dot\rho_q}{\partial \rho_q}
\end{bmatrix}
$$

$$
= \begin{bmatrix}
-r\rho_q^2 - 2r\rho_p\rho_q + (r - 1 - a)\rho_q - 2(a + 1)\rho_p + 1 - d & -r\rho_p^2 - 2r\rho_p\rho_q - (r + a + 1)\rho_p \\
\rho_q^2 + 2r\rho_p\rho_q + (d + a - r)\rho_p + 2a\rho_p + (d - r)\rho_q & r\rho_p^2 + (a - r + d)\rho_p + 2d\rho_q - d
\end{bmatrix}.
$$

Substituting $(0,0)$, $(1,0)$ and $(\rho_{p3}, \rho_{q3})$ into $J$, one can get the three Jacobian matrices $J_1$, $J_2$, $J_3$, at the respective fixed points (the values of $\rho_{p3}$ and $\rho_{q3}$ is numerically calculated for $J_3$) (the code for the numerical calculation of the Jacobian matrix is included in the electronic supplementary material). The determinants of $J_1$ and $J_2$ are negative, implying that $(0,0)$ and $(0,1)$ are saddle points.

The stability around $(\rho_{p3}, \rho_{q3})$ is more complicated. Let us denote the determinant and trace of $J_3$ as $\Delta$ and $\tau$. The discriminant $\tau^2 - 4\Delta$ is negative when $\rho_p$ is smaller and positive when $\rho_p$ is larger, indicating that the fixed point in low proliferation range is a spiral and in high proliferation range a node (figure 3a1). The trace $\tau$ is always negative indicating that $(\rho_{p3}, \rho_{q3})$ are stable spirals or nodes. Nevertheless, $\tau$ is closer to zero when the spiral locates at lower $\rho_p$ and higher $\rho_q$ ranges (figure 3a2), implying that the stability is rather fragile near this range. Moreover, the stable manifold of $(0,0)$ is $\rho_p = 0$ so that once the system hits $\rho_p = 0$, it will be absorbed into this wall until it reaches $(0,0)$.

Figure 3b shows the vector field under three parameter settings. The initial condition is $\rho_p(0) = 0.0001$ and $\rho_a(0) = 0$ to mimic the initial conditions in the Monte−Carlo simulations. Hence, the system starts from

[4]The rigorous analytical representations of $\rho_{p3}$ and $\rho_{q3}$ are too complicated to be shown here. For a clearer demonstration on their relation to $r$, $d$, $a$, we show the approximated representation.

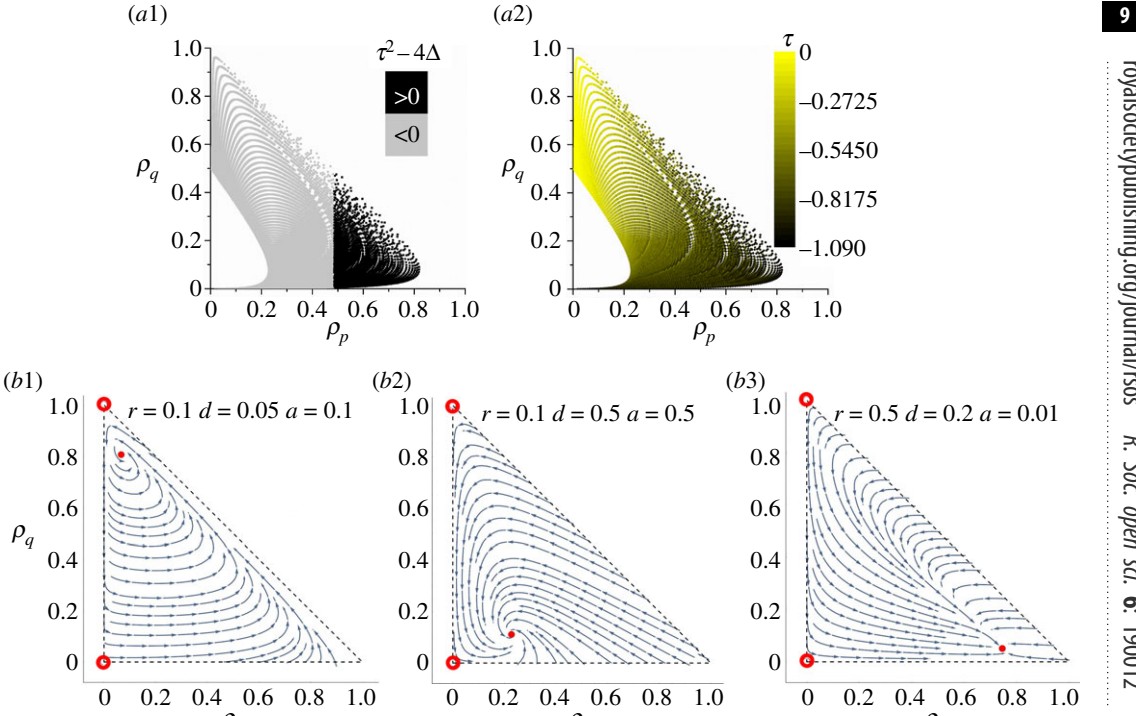

**Figure 3.** System dynamics under mean-field assumptions. The discriminant $\tau^2 - 4\Delta$ ($a$1) and trace $\tau$ ($a$2) of Jacobian at $(\rho_{p3}, \rho_{q3})$. ($b$) Vector fields for three different parameter settings. Note that the two saddles points at (0,0) and (0,1) are highlighted as red circles; in subfigures ($b$1) and ($b$2), the fixed points $(\rho_{p3}, \rho_{q3})$ are spirals, while in ($b$3) it is a node.

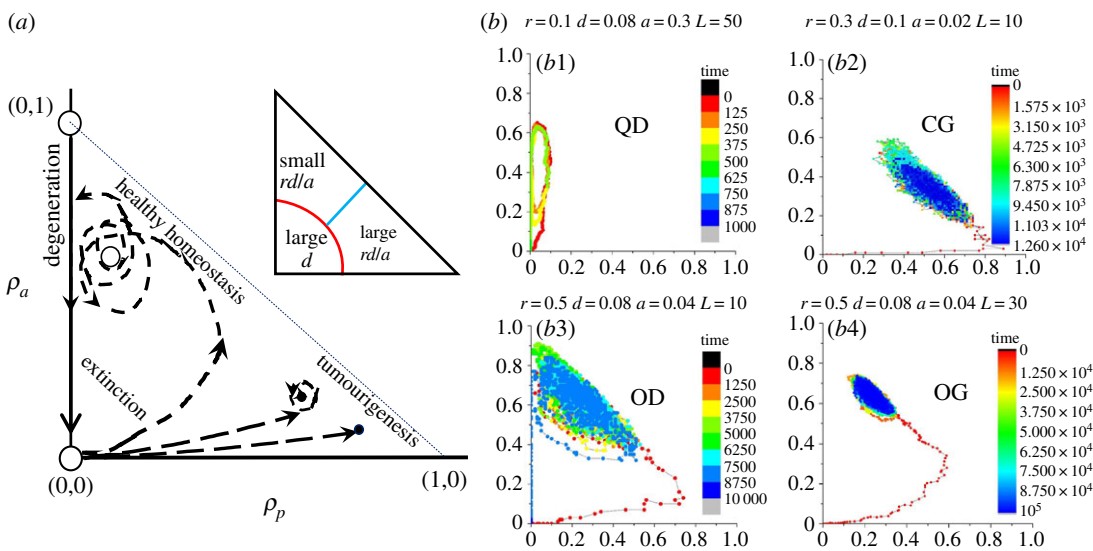

**Figure 4.** ($a$) System trajectories postulated from mean-field analysis. Inset: the schematic phase separation in relation to $r$, $d$, $a$ according to equation (3.4). ($b$) System trajectories from Monte–Carlo simulations: ($b$1) quick degeneration (QD), ($b$2) compact growth (CG), ($b$3) oscillatory degeneration (OD), ($b$4) oscillatory growth (OG). Colours stand for different ranges of time.

(0.0001, 0) and travels on the portrait along the arrows of the vector field until it approaches the spiral or node point, which is none other than the equilibrium or the so-called homeostasis of the system, and linear stability around these points indicates the robustness of this homeostasis. Under the mean-field assumptions (no fluctuation and infinite size), all systems with various parameter settings will find their equilibria without being absorbed into $\rho_p = 0$, therefore no extinctive or degenerative states exist. This discrepancy between the mean-field results and the Monte–Carlo outputs (figure 2) is caused by the neglect of the model's stochastic nature and the size-finiteness. Nevertheless, the actual system behaviours can still be postulated from the mean-field vector field as shown in figure 4:

(i) if the equilibrium state locates in the range of large $\rho_p$ and small $\rho_q$, i.e. a state showing tumorigenic features (the right-bottom corner), the system will stick to it with small random aberrations around the fixed point. The equilibria of compact growth should lie in this range;

(ii) if the equilibrium state locates in the range of small $\rho_p$ and large $\rho_q$, i.e. a state close to healthy homeostasis in most tissues (the top corner), the stability of this state is fragile and the system keeps swirling around the fixed point with large radii until it hits the absorbing wall $\rho_p = 0$ by chance and a degenerative process ensues. The patterns of oscillatory growth, oscillatory degeneration and quick degeneration should lie in this range; and

(iii) if the equilibrium state locates in the range of small $\rho_p$ and small $\rho_q$, i.e. a state unable to grow into a mature system (the left-bottom corner), it may also swirl around this fixed point before hitting $\rho_p = 0$ by chance and then die out. The extinctive patterns should lie in this range and the equilibria of scattered growth should exist narrowly on the boundary between the extinctive and the tumorigenic range.

The separation of the three major phases is controlled by a complicated function of $r$, $d$, $a$, derived from the analytical expression of $\rho_{p3}$ and $\rho_{q3}$. Yet, one could extract useful information from equations (3.4), that the phase separations heavily depend on two factors: $1 - d$ and $rd/a$ (see the inset plot of figure 4a), the former of which basically separates the extinctive and non-extinctive phases (red curve), while the latter separates healthy homeostasis and tumorigenesis (cyan line).

Sample trajectories from Monte–Carlo simulations are shown in figure 4b1–b4, for (b1) quick degeneration, (b2) compact growth, (b3) oscillatory degeneration and (b4) oscillatory growth. Note that (b3) and (b4) have the same $r$, $d$, $a$ but with different system size $L$, manifesting the effect of finite size in amplifying the oscillatory amplitude of dynamics (larger radii of orbits around fixed point in (b3)) and in reducing the system robustness (see the collapse of homeostasis in (b3)). These Monte–Carlo samples agree with our postulation based on mean-field analysis.

# 4. Phase diagram

So far, we have clarified why the diverse homeostatic patterns could be generally categorized into extinction, growth and degeneration based on mean-field analysis and the fixed point for healthy homeostasis is a spiral point with low robustness, rendering longtime oscillatory behaviours before an eventual collapse in the dynamics. For a thorough understanding on the physics of the model and its relation to some well-studied non-equilibrium processes, we will construct the complete phase diagram in the parameter space of $r$, $d$, $a$ based on statistical analysis of the Monte–Carlo results. The mechanism underlying the formation of all possible homeostatic patterns of various architectures, turnover rates and lifespans will be elucidated in the language of statistical mechanics.

## 4.1. Infinite growth

We first consider the phase diagram for an infinite space. The parameters for us to distinguish different phases are $\overline{N_p}(t)$ and $\overline{N_q}(t)$, which are the time evolutions of numbers of proliferative and quiescent cells averaged over $10^5$ sessions of simulations. To extrapolate the dynamics in infinite space from finite-size simulations, we do not need huge systems that consume enormous computational resources; instead, inspecting a short-time evolution in a small system is sufficient to imply the infinite-size system behaviours and to identify the boundaries between distinct phases.

Figure 5a shows the phase transition from extinctive phases to proliferative phases by decreasing $d$ for $r = a = 0$. The proliferative systems are characterized by the exponentially expanding proliferative cell number $\overline{N_p}(t)$; by contrast, extinctive systems fail to expand and the number of proliferative cells soon declines to zero. The critical value of $d$ that separates the two phases is around 0.62 in the case of $a = 0$ and $r = 0$ and the critical growth of $\overline{N_p}(t)$ is a power law with the critical exponent $\overline{N_p}(t)$ roughly 0.239.

Similarly, figure 5b shows the phase transition from proliferative phases to degenerative phases. The degenerative phase is characterized by the exponentially expanding cluster of quiescent cells with the proliferative cells scattered at the periphery of the cluster (figure 2g). Hence, $\overline{N_q}(t)/\overline{N_p}(t)$ is exponentially diverging in degenerative phases while keeping fluctuated in proliferative phases. The critical value of $d$ that separates two phases is 0.027 (red curve in figure 5) in the case of $a = 0.7$ and $r = 0$ with the power of the critical growth roughly 1.26.

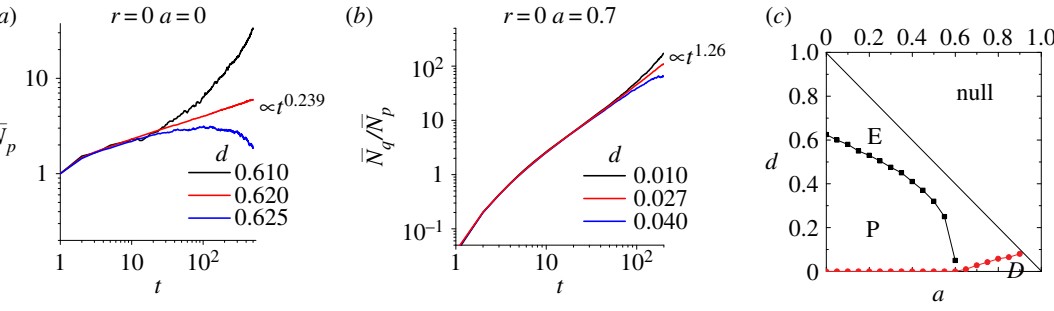

**Figure 5.** Phase separation for infinite space. (a) The separation of extinctive phases and proliferative phases by $\overline{N}_p(t)$. (b) The separation of proliferative phases and degenerative phases by $\overline{N}_q(t)/\overline{N}_p(t)$. (c) The phase diagram for infinite space at $r = 0$.

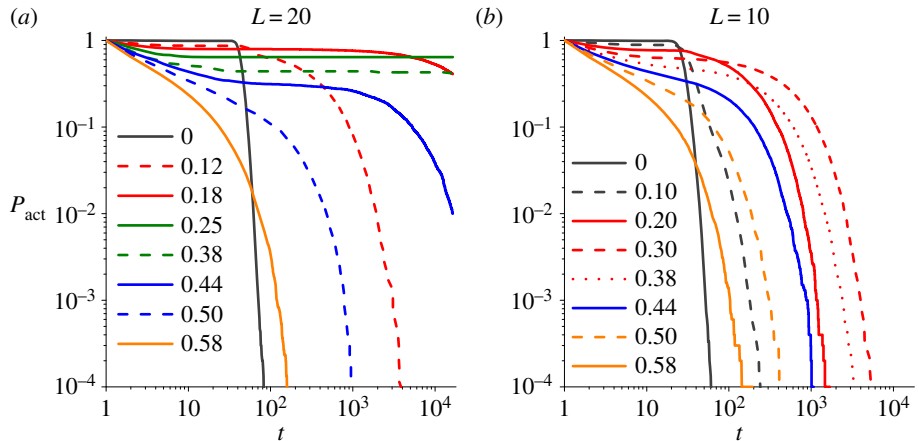

**Figure 6.** (a,b) The active probability $P_{\text{act}}(t)$ for varying $d$ with $r = 0.5$, $a = 0.3$. Different colours stand for different phases: grey for degeneration, green for proliferation, orange for extinction, red for longtime degeneration and blue for longtime extinction.

Scanning through the range of $r, d, a$ satisfying the constraints that keep each row in table 1 unity, one can identify all the critical values of $d$ for a certain set of $r$ and $a$. Figure 5c shows the phase diagram for infinite size at $r = 0$. Phase diagrams for infinite size at $r > 0$ can be similarly constructed but not shown here. One should notice that extinction occurs to large $d$ while degeneration occurs to small $d$. The proliferative phase exists in the medium range of $d$ and shrinks with increasing $a$ until it totally disappears when $a > 0.6$.

In contrast to the neat separation of extinctive, proliferative and degenerative phases by statistical analysis, the separation of different growth patterns (scattered, compact and oscillatory) is not feasible because of the absence of rigid criteria. Therefore, the identification of those growth patterns in the proliferative phase is only qualitative and the transitions among them have no clear critical boundaries. Section 4.2 will discuss in depth on this issue.

## 4.2. Finite growth

Next let us consider the effect of finite size on the emergence of more homeostatic patterns. Figure 6a shows the time evolution of statistical parameter $P_{\text{act}}(t)$ under system size $L = 20$, where $P_{\text{act}}(t)$ is the probability of the system with $\rho_p > 0$ (not being absorbed into degeneration or extinction) after a time $t$ and thus named as 'active probability'.[5] In finite systems, the proliferative phases, characterized as $P_{\text{act}}(t \to \infty) > 0$, shrink with decreasing $L$. In an extremely small system with $L = 10$ (figure 6b), the proliferative phase vanishes. By contrast, two new phases emerge and expand with decreasing $L$, one (blue curves) between extinctive and proliferative phases and the other (red curves) between degenerative and proliferative phases, with extremely long times to be absorbed into extinction or degeneration. These two phases correspond to slow extinction and oscillatory degeneration in figure

[5]A more extensively used parameter could be the survival probability, i.e. the probability of $\rho_p + \rho_q > 0$ in our model. Active probability is used here for the computational efficiency.

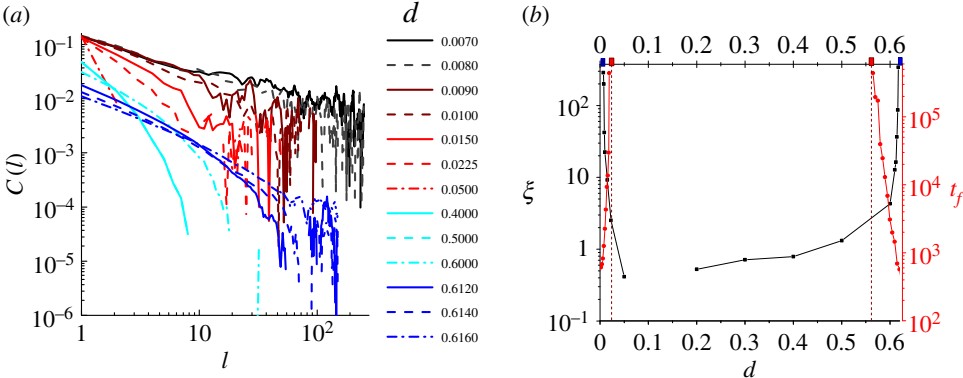

**Figure 7.** (a) Spatial correlation $C(l)$ measured at $r = 0$, $a = 0.01$ for different values of $d$. (b) Characteristic length of spatial correlation $\xi$ fitted from $C(l)$ (black) and characteristic lifespan $t_f$ fitted from the active probability $P_{\text{act}}(t)$ (red) for varying $d$.

2, and the active probabilities keep rather stable for a long time but eventually decay to zero. As is predicted from mean-field analysis, the finiteness of system size amplifies the stochasticity of the dynamics, thus causing proliferative systems near critical boundaries to collapse.

The loss of robustness owing to finite size could be addressed by the correlation length exceeding the system size. Although the interaction rules in our model are implemented on nearest neighbours, long-range correlations may appear when the parameters approach critical values. In statistical physics, correlation $C$ for a certain correlation distance $l$ and correlation time interval $T$ can be measured when a system reaches equilibrium as

$$C(l, T) = \langle \eta(x, t)\eta(x + l, t + T)\rangle - \langle \eta(x, t)\rangle\langle \eta(x + l, t + T)\rangle, \tag{4.1}$$

where $\eta(x, t)$ is the state of site $x$ at time $t$ and $\langle \cdot \rangle$ is the ensemble average, which is realized in simulation by taking the average over all $x$ in space, over a period of time after the system reaches equilibrium and over many sessions of simulations. For our purpose, we only need to scrutinize the spatial correlation of dynamics, hence setting $T$ as zero. To note, the equilibrium states do not exist in quick extinctive or degenerative phases, hence the correlation is only measurable for proliferative or quasi-proliferative phases.

Figure 7a shows the spatial correlation $C(l)$ from $10^4$ sessions of Monte–Carlo simulations for varying $d$ with $r = 0$, $a = 0.01$. It is observed that the decay of $C(l)$ approaches a power-law $l^{-\alpha}$ when $d$ is near 0 (black curves) or 0.6 (blue curves), whereas it approximates a quick exponential decay when $d$ lies between two ends (red and cyan curves). As in many physical systems [58,59], the correlation function $C(l)$ near critical states can be formulated by a multiplication of a power-law and an exponential decay, in a fashion as

$$C(l) \propto l^{-\alpha}\exp\left(\frac{-l}{\xi}\right), \tag{4.2}$$

for two-dimensional space, where $\xi$ is the characteristic correlation length. From equation (4.2), we know that when the correlation length $\xi$ goes to infinity, the spatial correlation decays exactly as $C(l) \propto l^{-\alpha}$. Therefore, we can first measure the values of $\alpha$ at two critical points, and then substitute these values into equation (4.2) to fit the correlation length $\xi$ for $d$ near those critical points. The two critical exponents are roughly 0.643 for smaller critical $d$ and 0.89 for the larger critical $d$. The fitted values of $\xi(d)$ with $r = 0$, $a = 0.01$ are shown in the black scattered line in figure 7b. Obviously, $\xi(d)$ diverges when $d$ approaches the two critical values (marked by blue rectangles on the top horizontal axis), whereas the intermediate range features very short correlation lengths.

Hence in an infinite space, the two critical points with infinite correlation lengths $\xi$ separate the phase space into three parts, the degenerative (small $d$) phase, the extinctive (large $d$) phase and the proliferative one with finite $\xi$ (intermediate $d$) (also see figure 5c). If the correlation length is far smaller than the space linear size $L$, i.e. $\xi(d) \ll L$, the finite-size effect could be ignored and the system behaves as in the infinite phase diagram, from which one can assert that the system is in the proliferative phase. By contrast, if $\xi(d) \gg L$, finite-size effect will dominate and the system falls into degeneration with a smaller $d$ or extinction with a larger $d$; therefore, the boundaries between degenerative or extinctive and proliferative phases move slightly owing to the size-finiteness.

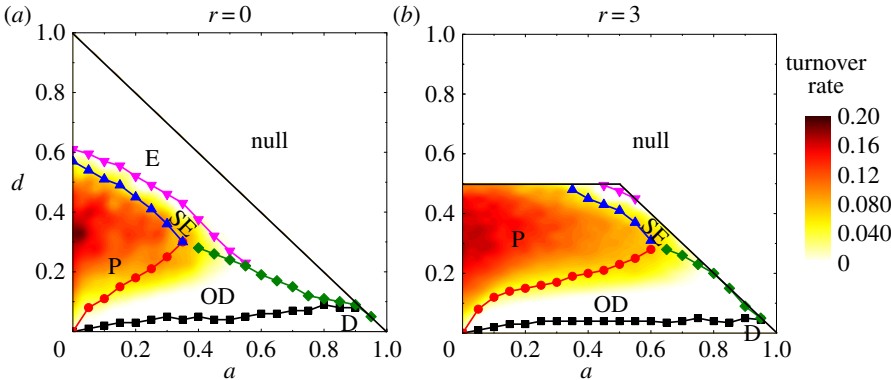

**Figure 8.** (a,b) Phase diagram for system with linear size $L = 20$. Note that the values of $r$, $d$, $a$ must meet the constraints that make each row in table 1 to be unity, hence any parameter setting violating these constraints would not be possible to give a result. Several phases are identified: extinction (E), proliferation including scattered growth, compact growth and oscillatory growth (P), slow extinction (SE), oscillatory degeneration (OD).

If $\xi(d) \sim L$, another two phases will emerge. The one between the proliferative and the extinctive phases is slow extinction, and that between the proliferative and the degenerative phases is oscillatory degeneration. At these phases, the characteristic timescale $t_f(d)$ of the decaying active probability $P_{act}(t)$ is supposed to diverge as $|d - d_c|^{\mu}$ where $d_c$ is the critical point separating the proliferative and the other two phases. Hence, $d_c$ can be extrapolated by fitting the algebraic divergence of $t_f(d)$. The red scattered line in figure 7b shows the divergence of $t_f(d)$ for $r = 0$, $a = 0.01$, $L = 20$, and two values of $d_c$ are marked by red rectangles on the top horizontal axis. As is seen in figure 7b, $t_f(d)$ and $\xi(d)$ separate the parameter space into five phases.

The phase diagram with $L = 20$ is exhibited in figure 8 for $r = 0$ and $r = 3$. The cell turnover rate is also shown in the plot in different colours. In proliferative phases, the turnover rates vary across scales from $10^{-3}$ to $10^{-1}$ and a low turnover rate corresponds to a larger correlation length $\xi$ and a high turnover rate to a shorter $\xi$. The compact growth in figure 2 is supposed to feature shortest correlation lengths and highest turnover rates, yet the change of turnover or correlation length is very smooth and therefore the transition between compact growth and scattered or oscillatory growth is continuous and the critical boundaries are not quantifiable.

# 5. Discussion

## 5.1. Towards specific biological problems

So far, we have explicated the formation of diverse homeostasis under a set of simple homogeneous rules with uniform parameters of $r$, $d$, $a$ through our simple SCA model. We have observed that the healthiest homeostasis which is characterized by a large composition of quiescent cells exists on the critical boundary between proliferative states and degenerative states and thus features quasi-stability owing to finite system size. Next, we concern the model's biological significance from the following two aspects.

Firstly, despite the large range for healthy homeostasis (oscillatory growth and oscillatory degeneration) observed in figure 8 for a small system, this range is indeed narrow for sufficiently large systems. Then a question naturally arises of how the systems can precisely tune the control parameters within those critical ranges. Secondly, many other biologically significant factors are absent in this model, among which are the long-range intercellular interactions (e.g. through chemotaxis), more crucial cell phenotypes (e.g. differentiated, migratory) and the spatio-temporal heterogeneity in the three key parameters ($r$, $d$, $a$) caused by factors such as specific tissue topologies, sophisticated metabolic regulations, the threshold dynamics of phenotype decision-making. There is extensive room for model extensions towards specific biological systems and the difference they make to the dynamics are of great importance. Here, we impose two tentative modifications on the system dynamics to briefly discuss their effects. Although these modifications are still very abstract, they may help us understand the behaviours of models under more realistic extensions.

Figure 9 shows a sample portrait of system trajectories under an oscillatory regulation on the death rate $d$ to model the impact from some exogenous temporal rhythms. The parameter setting is the same as

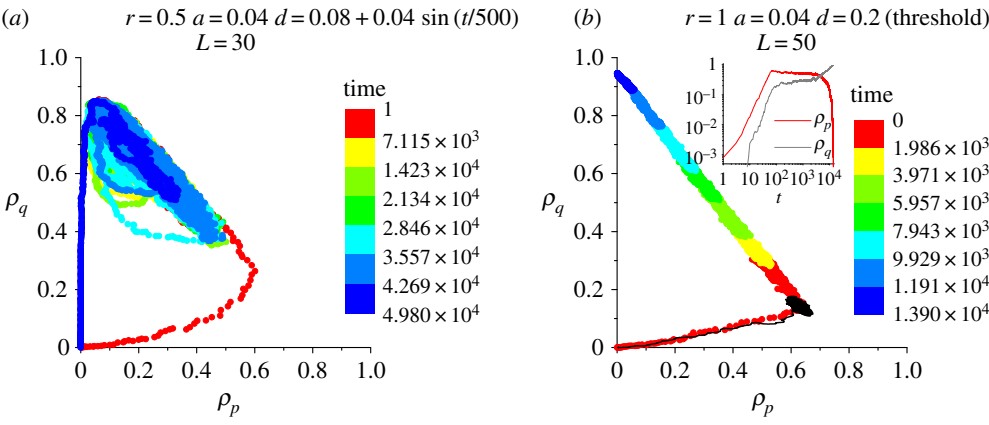

**Figure 9.** Sample system trajectories under (*a*) oscillatory regulation of death rate *d* and under (*b*) threshold implementation of cell death equivalent to a death rate *d*. The black line in (*b*) is the original system trajectory under non-threshold dynamics, which has an equilibrium state as compact growth. Inset in (*b*): the time evolution of $\rho_p(t)$ and $\rho_q(t)$ under the threshold implementation of cell death.

in figure 4*b*4. Under a constant $d = 0.08$, the system is supposed to reach healthy homeostasis with an infinite lifespan (oscillatory growth, see figure 4*b*4). By contrast, with an external oscillation of period 500 steps (roughly 120 days) on *d*, the robustness of this homeostasis wanes and eventually collapses after $5 \times 10^4$ steps (roughly 49 years).

Figure 9*b* shows a comparison between the original system trajectory (black) and the one under the threshold implementation of cell death (colour). With the threshold setting, cell death occurs only to a cell with an active lifespan exceeding $1/d$ time steps, where the active lifespan is defined as the total number of time steps the cell has experienced in a proliferative state. In theory, the implementation of threshold $1/d$ is statistically equivalent to the cell death with a background death probability *d*, yet the implementation of the threshold destroys the robustness of the original fixed point at the large $\rho_p$ and small $\rho_q$ in the long time. The system wanders around the fixed point at the early stage, followed by an incessant accumulation of quiescent cells, i.e. undergoing a process of systematic ageing (see the grey curve in the inset of figure 9*b*). The threshold implementation will also turn any point on $\rho_p = 0$ to be a fixed point, i.e. entailing an infinite number of absorbing states[6] to exist (in the model without threshold dynamics, (0,0) and (0,1) are the only two absorbing states).

Both examples of model extensions imply that the quasi-stable homeostasis could more extensively exist in the multicellular organizations than predicted by our simple SCA model if sophisticated biology sets in. Hence, the basic conclusions in §§3 and 4 from SCA still hold with the quasi-stable healthy homeostasis (oscillatory degeneration) phase spanning even more widely in the parameter space for more complicated extensions. Thus, the SCA model has captured the skeleton of multicellular homeostasis in real biology and the analysis of this model could also help understand the homeostasis and ageing of specific biological systems.

It should be noted that the essence of our model lies in the existence of the so-called absorbing states of the system. Robust homeostasis in theory corresponds to the fluctuated equilibrium states of the system; the ageing at the multicellular level, accordingly, is a dynamical transition of the system from the equilibrium to the absorbing states, which would be triggered and accelerated by factors such as finiteness of system size, turbulent physiological environment and the spatio-temporal heterogeneity originated from sophisticated biology.

## 5.2. Generalized epidemic process

Owing to the nature of cell division and contact inhibition, our model is reminiscent of some well-studied percolation models in non-equilibrium statistical physics, i.e. the so-called 'generalized epidemic process' (GEP) which originates from the epidemic spreading problem involving three basic state of individuals: susceptible, infected and immune [60,61]. In the GEP, the susceptible individuals may get infected through infected neighbours and the infected individuals can become resistant against diseases with a

---

[6]An absorbing state is a state which would never be escaped from once the system has reached it [37].

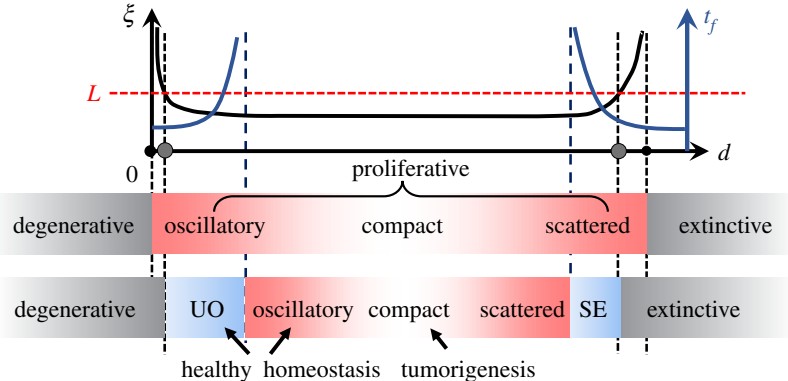

**Figure 10.** Summary of the physics under the formation of diverse homeostatic patterns and systematic ageing of healthy homeostasis. The vertical axis $\xi$ (left) is the characteristic correlation length (black) and $t_f$ (right) is the characteristic active duration, i.e. the lifespan of the dynamics (navy).

probability, and these settings have the same spirit as our models in that the susceptible individuals correspond to the vacant sites in our model, the infected ones to the proliferative cells and the immune ones to the quiescent cells. Also, many models in various fields share the same skeleton with the GEP, and hence the term 'generalized' indeed refers to a large family of models with modifications regarding the rules of transitions among the three states (see reviews in [60–62]). These models belong to the GEP family in the sense that they can be described by a general form of a Langevin equation for individual density $\rho(x, t)$ at position $x$ and time $t$:

$$\frac{\partial \rho(x, t)}{\partial t} = a \underbrace{\nabla^2 \rho}_{\text{diffusion}} + b \underbrace{R\left(\rho, \int_{-\infty}^{t} \rho(x, t')\,\mathrm{d}t'\right)\rho}_{\text{memory-dependent growth}} + \underbrace{\zeta\left(\rho, \int_{-\infty}^{t} \rho\,\mathrm{d}t', L, \ldots\right)}_{\text{Gaussian noise}}, \tag{5.1}$$

indicating that their statistical behaviours have many common features related to the universality class of directed percolation and dynamical percolation (see [37,63] for a thorough introduction on the non-equilibrium universality classes). The role of the immune individuals, i.e. the quiescent cells in our model, is to provide a mechanism to store local memory in the system, which is supposed to be essential to 'stabilize' the system as well as to introduce the absorbing states other than $\rho \equiv 0$. The noise can have several origins such as the endogenous stochasticity of state-updating, the exogenous random disorder imposed on the system. The structure of the noise decides how the system evolves into those absorbing states.

Based on our knowledge about the GEP, transitions between pandemic (proliferative) and endemic (extinctive and degenerative) phases generally belong to dynamical percolation which crosses over to directed percolation if the extent of immunization (cell-cycle arrest) is reduced and the first-order transitions may also conditionally occur between a pandemic and a compact endemic (degenerative) phase. The criticality of our SCA model with biology-relevant extensions should be scrutinized in future studies to solidify our non-equilibrium phase transition theory of multicellular homeostasis and ageing.

# 6. Conclusion

In this study, we conceive a simple three-state SCA with homogeneous state-updating rules based on fundamental cell biology. The model reproduces the diverse homeostatic states in respect of multicellular architecture, dynamics of subpopulation, turnover rate and lifespan, as observed in previous complicated multiscale models and experiments. Through mean-field analysis, we find that the system has three basic phases, namely the proliferative (including tumourigenic and healthy homeostasis), the degenerative, and the extinctive phase. Healthy homeostasis innately has low robustness and it may turn degenerative over time with a large spectrum of lifespan, while the tumourigenic homeostasis is extremely robust with an infinite lifespan. For a finite system, we plot the phase diagrams of the model based on statistical analyses and clarify the physics underlying the emergence of the basic three phases along with the other two phases where quasi-stable healthy homeostasis exists (figure 10). These results help us relate the model to the well-studied generalized

epidemic process and confirm that the systematic ageing of healthy homeostasis occurs as a dynamical transition from fluctuating equilibrium states to an absorbing state (which is the degenerative state in our model). These transitions are fluctuation-driven and manifested with finite size of space. We also discuss some biology-relevant extensions of the models and find that the robustness of healthy homeostasis would become more fragile owing to the spatio-temporal heterogeneity introduced by the more sophisticated biology. A hypothesis is hence proposed that whatever other biological details are introduced as in many complicated models, the formation of homeostasis in such systems could be understood under the theoretical framework of our simple model and of the generalized epidemic processes. Furthermore, the ageing of the healthy homeostasis with a finite lifespan would exist more extensively in the parameter space if more sophisticated biology is considered. Accordingly, a non-equilibrium phase transition theory might be established in future for the homeostasis as well as ageing of multicellular organizations and its applicability to more specific biological problems of tissue modulation, cancer prevention or anti-ageing issues could be one of any upcoming research themes.

Data accessibility. Our C++ codes of the Monte–Carlo simulations for the statistical analysis and of the numerical simulations for the stability analysis have been uploaded as part of the electronic supplementary material.

Authors' contributions. Y.L. and Y.C. designed the study. Y.L. established the model and analysed the output through mean-field analysis and statistical methods. Y.L. and A.C. performed numerical and Monte-Carlo simulations. Y.L., A.C. and Y.E. plotted phase diagrams. Y.L. interpreted the results and wrote the manuscript. Y.L. and Y.C. refined and revised the manuscript. All authors gave final approval for publication.

Competing interests. We declare we have no competing interests.

Funding. This work is supported by Japan Society for the Promotion of Science.

Acknowledgements. We are sincerely thankful to Didier Sornette, Fernando Peruani, Bastien Chopard, Hiroki Yamaguchi for having inspiring discussions with us. We also appreciate the valuable comments from the anonymous reviewers.

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
