## [Reviewer comments · Royal Society Open Science]

Review History

RSOS-190012.R0 (Original submission)

Review form: Reviewer 1 (Chay Giles Blair Paterson)

Is the manuscript scientifically sound in its present form?

Yes

Are the interpretations and conclusions justified by the results?

Yes

Is the language acceptable?

Yes

Is it clear how to access all supporting data?

Yes

Do you have any ethical concerns with this paper?

No

Have you any concerns about statistical analyses in this paper?

No

Recommendation?

Accept with minor revision (please list in comments)

Comments to the Author(s)

I enjoyed this paper, and think it is a sound approach to an interesting set of scientific problems.

One thing that caught my eye was that the phase portraits and system trajectories in section 3 were strongly reminiscent of this earlier work, which was not discussed or cited:

Warren, P.B., 2009. Cells, cancer, and rare events: Homeostatic metastability in stochastic nonlinear dynamical models of skin cell proliferation. *Physical Review E*, 80(3), p.030903.

In particular, this work derives similar phase portraits and sketches of the system dynamics, including tumorigenesis as an attractor of biological systems that escape some basin of stability, representing homeostasis. While the current paper by Y. Lou et al. is a more substantial and detailed study of the same problems, I feel this paper would be enriched by a brief (2 sentence?) discussion of P.B. Warren 2009.

Otherwise I think their survey of the literature seems fine. The essential papers by D. Drasdo and B.D. Simons are all there.

The introduction of the model could also be described a bit more clearly. Something like "Cells switch at a rate proportional to the number of neighbours of type X" would help a lot. At present, the simple description is very short and the abstract mathematical description much longer, which has the effect of swamping the simple explanation. It is a simple and elegant model and I think this should be emphasised.

Review form: Reviewer 2

Is the manuscript scientifically sound in its present form?

Yes

Are the interpretations and conclusions justified by the results?

Yes

Is the language acceptable?

Yes

Is it clear how to access all supporting data?

Yes

Do you have any ethical concerns with this paper?

No

Have you any concerns about statistical analyses in this paper?

No

Recommendation?

Accept with minor revision (please list in comments)

Comments to the Author(s)

Yuting et al present a model of multicellular homeostasis using a 2D stochastic cellular automaton framework. Although quite technical in places all the methods, including the master equations and mean field approximation are well presented and the assumptions made are appropriate. Overall, the development of the model is well explained and the exploration of the phase planes both statistically (Monte Carlo) and analytically (Mean Fields followed by Jacobian) is well done. The sharp transitions seen in the phase planes are not that surprising. Towards the end of the paper (Figure 9), the authors introduce realism into the model by including a threshold-dependent cell death they although the oscillating value for cell death is a little odd.

One point that I found unclear was the technical details of the correlation length (method behind Figure 7) but it makes sense that the agents will be highly correlated at the start of the simulation and at the end of the simulation if they undergo a quick extinction or a degradation as they seem to suggest.

Decision letter (RSOS-190012.R0)

28-May-2019

Dear Ms Lou

On behalf of the Editors, I am pleased to inform you that your Manuscript RSOS-190012 entitled "Homeostasis and systematic aging as nonequilibrium phase transitions in computational multicellular organizations" has been accepted for publication in Royal Society Open Science subject to minor revision in accordance with the referee suggestions. Please find the referees' comments at the end of this email.

The reviewers and handling editors have recommended publication, but also suggest some minor revisions to your manuscript. Therefore, I invite you to respond to the comments and revise your manuscript.

- Ethics statement

- Data accessibility

It is a condition of publication that all supporting data are made available either as supplementary information or preferably in a suitable permanent repository. The data accessibility section should state where the article's supporting data can be accessed. This section should also include details, where possible of where to access other relevant research materials such as statistical tools, protocols, software etc can be accessed. If the data has been deposited in an external repository this section should list the database, accession number and link to the DOI for all data from the article that has been made publicly available. Data sets that have been

deposited in an external repository and have a DOI should also be appropriately cited in the manuscript and included in the reference list.

If you wish to submit your supporting data or code to Dryad (<http://datadryad.org/>), or modify your current submission to dryad, please use the following link:
<http://datadryad.org/submit?journalID=RSOS&manu=RSOS-190012>

- **Competing interests**

- **Authors' contributions**

- **Acknowledgements**

- **Funding statement**

Because the schedule for publication is very tight, it is a condition of publication that you submit the revised version of your manuscript before 06-Jun-2019. Please note that the revision deadline will expire at 00.00am on this date. If you do not think you will be able to meet this date please let me know immediately.

When submitting your revised manuscript, you will be able to respond to the comments made by

the referees and upload a file "Response to Referees" in "Section 6 - File Upload". You can use this to document any changes you make to the original manuscript. In order to expedite the processing of the revised manuscript, please be as specific as possible in your response to the referees. We strongly recommend uploading two versions of your revised manuscript:

If your manuscript is newly submitted and subsequently accepted for publication, you will be asked to pay the article processing charge, unless you request a waiver and this is approved by Royal Society Publishing. You can find out more about the charges at <http://rsos.royalsocietypublishing.org/page/charges>. Should you have any queries, please contact opscience@royalsociety.org.

Kind regards,
Andrew Dunn

Royal Society Open Science Editorial Office
Royal Society Open Science
openscience@royalsociety.org

on behalf of Prof Pietro Cicuta (Subject Editor)
openscience@royalsociety.org

Editor Comments to Author (Dr Pietro Cicuta):

Associate Editor: 1

Comments to the Author:

The reviewers make a few small but relevant suggestions for improvement.

Editor: 2

Comments to the Author:

Sending to peer review.

Reviewer comments to Author:

Reviewer: 1

Comments to the Author(s)

I enjoyed this paper, and think it is a sound approach to an interesting set of scientific problems.

One thing that caught my eye was that the phase portraits and system trajectories in section 3 were strongly reminiscent of this earlier work, which was not discussed or cited:

Warren, P.B., 2009. Cells, cancer, and rare events: Homeostatic metastability in stochastic nonlinear dynamical models of skin cell proliferation. *Physical Review E*, 80(3), p.030903.

In particular, this work derives similar phase portraits and sketches of the system dynamics, including tumorigenesis as an attractor of biological systems that escape some basin of stability, representing homeostasis. While the current paper by Y. Lou et al. is a more substantial and detailed study of the same problems, I feel this paper would be enriched by a brief (2 sentence?) discussion of P.B. Warren 2009.

Otherwise I think their survey of the literature seems fine. The essential papers by D. Drasdo and B.D. Simons are all there.

The introduction of the model could also be described a bit more clearly. Something like "Cells switch at a rate proportional to the number of neighbours of type X" would help a lot. At present, the simple description is very short and the abstract mathematical description much longer, which has the effect of swamping the simple explanation. It is a simple and elegant model and I think this should be emphasised.

Reviewer: 2

Comments to the Author(s)

Yuting et al present a model of multicellular homeostasis using a 2D stochastic cellular automaton framework. Although quite technical in places all the methods, including the master equations and mean field approximation are well presented and the assumptions made are appropriate. Overall, the development of the model is well explained and the exploration of the phase planes both statistically (Monte Carlo) and analytically (Mean Fields followed by Jacobian) is well done. The sharp transitions seen in the phase planes are not that surprising. Towards the

end of the paper (Figure 9), the authors introduce realism into the model by including a threshold-dependent cell death they although the oscillating value for cell death is a little odd.

One point that I found unclear was the technical details of the correlation length (method behind Figure 7) but it makes sense that the agents will be highly correlated at the start of the simulation and at the end of the simulation if they undergo a quick extinction or a degradation as they seem to suggest.

Author's Response to Decision Letter for (RSOS-190012.R0)

See Appendix A.

Decision letter (RSOS-190012.R1)

04-Jun-2019

Dear Ms Lou,

I am pleased to inform you that your manuscript entitled "Homeostasis and systematic aging as nonequilibrium phase transitions in computational multicellular organizations" is now accepted for publication in Royal Society Open Science.

on behalf of Dr Pietro Cicuta (Subject Editor)
openscience@royalsociety.org

Appendix A

Response Letter

Yuting Lou
30th May, 2019

Dear editor of Royal Society Open Science,

We thank the editors and reviewer's comments on this manuscript and complete minor revisions on it. We have added some reference papers to the introduction part and revised the description of our model in a more straightforward manner (highlighted in yellow). Besides, we added some technical demonstrations on how to obtain the correlation length in Sec. 4.2 (highlighted in pink). We hope that these revisions improve the quality of this paper. The detailed answers to each point by the reviewers are listed as below.

Reviewer 1 (correspondent revision is highlighted in yellow in the manuscript):

1. On the reference of P.B. Warren 2009

One thing that caught my eye was that the phase portraits and system trajectories in section 3 were strongly reminiscent of this earlier work, which was not discussed or cited:

Warren, P.B., 2009. Cells, cancer, and rare events: Homeostatic metastability in stochastic nonlinear dynamical models of skin cell proliferation. *Physical Review E*, 80(3), p.030903.

In particular, this work derives similar phase portraits and sketches of the system dynamics, including tumorigenesis as an attractor of biological systems that escape some basin of stability, representing homeostasis. While the current paper by Y. Lou et al. is a more substantial and detailed study of the same problems, I feel this paper would be enriched by a brief (2 sentence?) discussion of P.B. Warren 2009.

Answer:

We thank the reviewer for the recommendation of this paper. It is indeed very important and relevant to our research and so are the other related papers therein. We have added several sentences in the introduction as suggested (line 37-42) to enrich the background part.

2. On the description of model in Introduction

The introduction of the model could also be described a bit more clearly. Something like "Cells switch at a rate proportional to the number of neighbours of type X" would help a lot. At present, the simple description is very short and the abstract mathematical description much longer, which has the effect of swamping the simple explanation. It is a simple and elegant model and I think this should be emphasised.

Answer:

We take the suggestion here and try to make model description in Introduction easier to read (line 52-54). We thank the reviewer's appreciation on our model.

.....

Reviewer 2 (correspondent revisions highlighted in pink in the manuscript)

1. On the odd types of cell death

The authors introduce realism into the model by including a threshold-dependent cell death they although the oscillating value for cell death is a little odd.

Answer:

We are sorry for unclear demonstration on the motivation of introducing these modifications. These may not have accurate biological meaning and we propose them here only to demonstrate the model behaviour under some non-uniform and constant setting of the control parameters and provide some hints for more realistic extensions. We add some explanations on our motivation in line 494-495.

2. On the technical details of the correlation length

One point that I found unclear was the technical details of the correlation length (method behind Figure 7) but it makes sense that the agents will be highly correlated at the start of the simulation and at the end of the simulation if they undergo a quick extinction or a degradation as they seem to suggest.

Answer:

We thank the reviewer for pointing out this problem of unclear demonstration on correlation length. The correlation length is calculated by fitting parameters in the correlation function (eq.10). The average $\langle \rangle$ is taken over a period of homeostatic growth (where the numbers of proliferative and quiescent cells are fluctuating around some values before the system goes to extinct or degenerate) and over 40000 sessions of simulations. A very important thing is that correlation length is only measurable when the system is in the proliferative phase with infinite size (see the phase P in Fig.5(c)).

The details of fitting procedures is as such: 1) by varying the control parameter d , one can find the critical points where the correlation function $C(l)$ decays in a power-law function $C(l) \propto l^{-\alpha}$. 2) Measure the powers α of decay at the critical points. 3) Calculate the correlation function $C(l)$ for the range of control parameter near this critical point and fit Eq.10 to obtain the correlation length ξ .

We make the technical demonstration on correlation length clearer in the Sec. (line 421-423 and line 431-435) and hope this revision may improve the readability of this part.